# BMI Modifies Increased Mortality Risk of Post-PCI STEMI Patients with AKI

**DOI:** 10.3390/jcm11206104

**Published:** 2022-10-17

**Authors:** Reut Schvartz, Lior Lupu, Shir Frydman, Shmuel Banai, Yacov Shacham, Amir Gal-Oz

**Affiliations:** 1Department of Anesthesia Pain and Intensive Care, Tel Aviv Sourasky Medical Center, Tel Aviv 62431, Israel; 2Department of Cardiology, Tel Aviv Sourasky Medical Center, Tel Aviv 62431, Israel; 3Sackler Faculty of Medicine, Tel Aviv University, Tel Aviv 39040, Israel; 4Cardiac Intensive Care Unit, Tel Aviv Sourasky Medical Center, Tel Aviv 62431, Israel

**Keywords:** STEMI, PCI, AKI, obesity paradox

## Abstract

Mortality from acute ST elevation myocardial infarction (STEMI) was significantly reduced with the introduction of percutaneous catheterization intervention (PCI) but remains high in patients who develop acute kidney injury (AKI). Previous studies found overweight to be protective from mortality in patients suffering from STEMI and AKI separately but not as they occur concurrently. This study aimed to establish the relationship between AKI and mortality in STEMI patients after PCI and whether body mass index (BMI) has a protective impact. Between January 2008 and June 2016, two thousand one hundred and forty-one patients with STEMI underwent PCI and were admitted to the Tel Aviv Medical Center Cardiac Intensive Care Unit. Their demographic, laboratory, and clinical data were collected and analyzed. We compared all-cause mortality in patients who developed AKI after PCI for STEMI and those who did not. In total, 178 patients (10%) developed AKI and had higher mortality (*p* < 0.001). Logistic regression analysis was performed to determine the relationship between AKI, BMI, and mortality. AKI was significantly associated with both 30-day and overall mortality, while BMI had a significant protective effect. Survival analysis found a significant difference in 30-day and overall survival between patients with and without AKI with a significant protective effect of BMI on survival at 30 days. AKI presents a major risk for mortality and poor survival after PCI for STEMI, yet a beneficial effect of increased BMI modifies it.

## 1. Introduction

Ischemic heart disease (IHD) is the leading cause of mortality [1]. The most severe manifestation of IHD is myocardial infarction (MI). The gold standard for the management of ST elevation MI (STEMI) is percutaneous catheterization intervention (PCI) which decreases mortality dramatically [2].

Many STEMI patients suffer from a deterioration in renal function [3]; acute kidney injury (AKI) occurs in up to 13% of patients with acute MI [4] and raises the risk for both early and late mortality [5,6]. STEMI patients after PC are especially at risk for AKI development after PCI as contrast medium is a significant contributor [7]. Other risk factors include impaired renal function, reduced ejection fraction (EF), and elevated C-reactive protein (CRP) [8,9].

Body mass index (BMI) strongly predicts morbidity and mortality in STEMI [10,11]. Median survival is reduced by BMI class [12,13], yet it has a nonlinear effect; mortality is lower in elevated BMI patients and higher in very low and very high BMI [14,15]. Despite the higher incidence of morbidity, this phenomenon of obese patients with better outcomes is termed ‘The Obesity Paradox’ and is observed in many clinical scenarios [13,16]. Regarding AKI development, obese patients are at increased risk, and its incidence increases with each BMI class [17].

AKI and obesity are independent risk factors for mortality in STEMI patients with different effects. Whether the Obesity Paradox persists in this unique setting, as they occur in the same patient, is still unanswered. This study aimed to investigate whether BMI modifies the relationship between AKI and mortality among a cohort of STEMI patients treated with PCI.

## 2. Materials and Methods

### 2.1. Population

We performed a retrospective analysis of all STEMI patients treated with PCI admitted to the Cardiac Intensive Care Unit (CICU) at the Tel-Aviv Medical Center between 1 January 2008 and 31 May 2016; the follow-up period ended on 6 June 2017. Data were collected from the electronic medical records and included demographics, BMI (self-reported height and weight), smoking status, comorbidities (hypertension (HTN), diabetes mellitus (DM), past MI, and hyperlipidemia (HPL)), laboratory tests (creatinine (Cr), Hb, C-reactive protein (CRP), and creatine phosphokinase (CPK)). All patients underwent an echocardiographic screening examination within three days of admission to assess left ventricular (LV) ejection fraction (EF); clinical data included door to balloon time, time to reperfusion, contrast volume infused, significant coronary artery disease (CAD, defined as lumen narrowing >70% during PCI), hemodynamic instability (recorded use of inotrope or intra-aortic balloon pump insertion), blood products administration, complications, and mortality.

### 2.2. Data Collection and Definition

The study was approved by the institutional ethics committee. The medical records of all patients who suffered from STEMI and underwent PCI during the study period were reviewed. Data on kidney function were collected from routine laboratory records. AKI was defined using the KDIGO criteria [18] as an increase in serum creatinine by 0.3 mg/dL or more within 48 h compared with admission creatinine level or an increase in serum creatinine above 1.5 times baseline within seven days of admission. BMI was calculated from height and weight (self-reported) recorded at ICU admission (BMI = weight [kg]/height [m^2^]). The primary outcome was all-cause mortality on 30 days post PCI and during the study follow-up period. Assessment of survival after hospital discharge was determined from computerized records of the population registry bureau and was available for all included patients until the end of the study period on 6 June 2017.

### 2.3. Statistical Analysis

Continuous variables were reported as means and standard deviation (SD) for normally distributed, and median and interquartile range (IQR) for non-normal, categorical variables were reported as percentages. Univariate analysis was performed with parametric and nonparametric tests as appropriate. AKI was defined as a dichotomous variable, and BMI as a continuous variable. Logistic regression was used for multivariant analysis. Predictors related to mortality on univariate analysis (*p* < 0.05) were entered into the model. BMI, HPL, and CPK were forced into the model due to their clinical relevance to the development of AKI and mortality.

All potential confounders are listed in Table 1. Interactions between AKI, BMI, and other covariates were tested with the addition of an interaction term, significant multiple interactions (*p* < 0.05) were found, and the interaction terms were included in the final regression models. A logistic regression model was performed to assess the association between AKI, BMI, and other potential confounders for mortality at 30 days post PCI and overall mortality at the end of the follow-up period. The Cox regression model was performed to assess survival at these periods, with the cohort divided according to AKI incidence to evaluate the contribution of BMI, and other potential confounders.

Analyses were performed on SPSS version 25 (IBM corporation, Armonk, NY, USA).

## 3. Results

### 3.1. Baseline Characteristics

Two thousand one hundred forty-one patients underwent PCI due to STEMI during the study period. In total, 178 (10%) suffered from AKI (N = 1706 with a recorded outcome), and 1045 (65%) were overweight (BMI > 25) (N = 1605 with a recorded BMI).

The baseline clinical characteristics of patients with and without AKI are shown in Table 1. Patients who suffered AKI were older (70.8 yrs vs. 60.4 yrs, *p* < 0.001), the proportion of females among them was higher (27% vs. 18.7%, *p* = 0.009), and the proportion of smokers was lower (33% vs. 52.1% *p* < 0.001). There was no difference in BMI between the groups. There was a higher prevalence of DM (32% vs. 20.5% *p* < 0.001) and HTN (69% vs. 40.1% *p* < 0.001) with no difference in the proportion of HPL prevalence among the AKI group. On admission patients with AKI had a higher creatinine level (1.3 mg/dL vs. 1.1 mg/dL *p* < 0.001), CRP (9.3 mg/L vs. 3.8 mg/L *p* < 0.001) and CPK (918 U/L vs. 776 U/L *p* < 0.001), but lower Hb (13.6 mg/dL vs. 14.4 mg/dL a long time to reperfusion (232.5 min vs. 165 min, *p* = 0.001), and the proportion of hemodynamically unstable patients was higher (16.9% vs. 2.2%, *p* < 0.001), they had a higher proportion of multiple vessel disease (more than two stenotic vessels 63.5% vs. 54.7% *p* = 0.008) and their EF after PCI was lower (45% vs. 50%, *p* < 0.001). Patients who did not have a recorded creatinine and AKI diagnosis were not included in the analysis. A subgroup analysis was performed to establish whether patients with missing data had different baseline characteristics than those who were included in the final analysis; the results are shown in Appendix A.

The baseline clinical characteristics of patients with and without overweight (BMI > 25) are shown in Table 2. Most of our cohort were overweight and class I obese (BMI Class frequencies within patients with and without AKI are shown in Appendix A).

Overweight patients were younger (60 yrs vs. 64 yrs *p* < 0.001), and the proportion of females among them was higher (58.1% vs. 41.9%, *p* = 0.005) with no difference in the proportion of smokers. There was a higher prevalence of DM (26.7% vs. 17.1%, *p* < 0.001), HTN (45% vs. 37.9%, *p* = 0.006) and HPL (49% vs. 41.4%, *p* = 0.004) among the overweight group. On admission, patients with overweight had no difference in creatinine level but higher CRP (5.4 mg/L vs. 3 mg/L *p* < 0.001) and Hb (14.4 g/dL vs. 14 g/dL *p* = 0.001) with no difference in CPK. There was no difference in the amount of contrast volume infused, time to reperfusion, and hemodynamic instability recorded; despite a higher proportion of CAD (59.8% vs. 52.9%, *p* = 0.001), their EF after PCI was higher (45.9% vs. 45%, *p* < 0.001). There was no difference in the incidence of AKI between overweight and non-BMI were not included in the analysis. A subgroup analysis was performed to establish whether patients with missing data had different baseline characteristics than those who were included in the final analysis; the results are shown in Appendix A.

### 3.2. Primary Outcome

Two hundred fifty patients have died (11.7% of 2125 patients with a recorded outcome). Both mortality at 30 days and mortality at the end of the follow-up period (overall mortality) were significantly higher among patients who suffered AKI (14.9% vs. 1.3%, *p* < 0.001, N = 1678, and 38.2% vs. 9%, *p* < 0.001, N = 1706, respectively).

Among overweight and non-overweight patients, there was no difference in mortality at 30 days (2.7% vs. 4.1% *p* = 0.11), but the overall mortality was significantly lower (9% vs. 13.9% *p* = 0.002) in the overweight group (Figure 1).

When BMI was categorized according to WHO clinical class, a significant difference between classes was found in overall mortality (*p* = 0.02), the highest proportion of deaths was among the lower BMI groups (28.5% and 13.6% BMI < 18.5, 18.6–24.9, respectively), and the lowest proportion was observed among the obese (8.5% and 7% BMI 30–34.9, BMI 35–39.9, respectively) (Table 3).

### 3.3. Multivariant Analysis

AKI development was found to be a significant risk factor for both mortality at 30 days (adjOR = 41.5, CI 95% 3.9–431.1, *p* = 0.002) and overall mortality (adjOR = 2.9, CI 95% 1.2–6.9, *p* = 0.01). After adjusting for predictors of significant coronary and metabolic disease in the multivariant model, BMI had a significant protective effect on mortality at 30 days (adjOR = 0.40, CI 95% 0.21–0.75, *p* = 0.005) and a modest significant effect on overall mortality (adjOR = 0.91, CI 95% 0.84–0.99, *p* = 0.04). The most significant contributor to mortality risk both at 30 days and overall was hemodynamic instability (adjOR = 247.0, CI 95% 8.1–7512.7, *p* = 0.09, and adjOR = 23.7, CI 95% 5.1–108.2, *p* < 0.001, respectively) (Table 4).

A survival analysis model was performed with the cohort divided into stratum relative to AKI status. BMI had a significant protective effect on survival at 30 days (adjHR = 0.77, CI 95% 0.64–0.94, *p* = 0.01) (Table 5).

## 4. Discussion

This study examined the effect BMI has on mortality among STEMI patients who underwent PCI and suffered from AKI. We found AKI was an independent risk factor for both 30-day and overall mortality post PCI. Previous studies established this increased risk for mortality after STEMI by 2–12 times [3,5,19] and reduced long-term survival by at least 20 times [20]. Deterioration in renal function is more likely in older, female patients, with comorbidities and severe coronary disease [21]; these characteristics were demonstrated in our cohort as well.

Obesity is a known risk factor for AKI development in critically ill patients [17]. The impact of BMI on mortality in critically ill patients with AKI follows the classical U-shaped pattern. The mechanism remains unknown, yet speculations regarding vasoconstriction and its effect on renal tissue are under investigation [22], and a possible protective effect of overweight linked to the catabolic state of these patients was proposed [23]. In our study, AKI incidence was not significantly higher in obese patients.

The significant finding of this study is that patients who developed AKI had gained a survival benefit from increased BMI, despite having significant contributing negative factors for a worse outcome, such as a long time to reperfusion, higher proportion of multiple vessel disease, an increased need for hemodynamic support, and lower EF after PCI. A possible explanation is that most of our cohort included overweight and class I obese with few morbidly obese patients.

This emphasizes that the interplay between BMI and AKI is poorly defined and not as straightforward in this complex setting. A different study attempted to establish a reversed link between change in renal function affecting the relationship of BMI and mortality after MI; they found overweight is protective against mortality in reduced renal function yet less among patients with very low renal function [24].

One of the main pathways investigated for a possible link between obesity and outcomes of pathophysiological processes, MI included, is the effect of inflammation. The inflammatory response stands at the core of the pathophysiology of myocardial injury [25] and is also a major component enhancing renal cellular destruction after AKI [26].

Overweight and obese individuals have a different cytokine profile in their plasma [27,28], and STEMI patients with visceral obesity were found to have elevated pro-inflammatory and decreased anti-inflammatory interleukins (IL) levels [25] originating from the adipose tissue, which secretes adiponectin and leptin, which are potential cytokines [27,28].

Low adiponectin was a risk factor for cardiovascular diseases and related to higher markers of cardiac injury [29]. Adiponectin induces the production of anti-inflammatory cytokines, impairs the production of the pro-inflammatory cytokine and affects leukocyte differentiation [30]. Leptin can induce the production of pro-inflammatory cytokines such as TNF α, IL-6, and 12 [31,32]. Increased levels of IL-6 stimulate the liver to synthesize and secrete CRP [31], which functions as a pro-atherosclerotic factor [33] and serves as a risk marker for mortality from acute coronary syndrome (ACS) after 30–90 days [25,34]. Our study did control for CRP as an inflammatory marker, yet while its significance was clear, its added risk was minor. Most of our cohort were overweight and not morbidly obese, which might be the reason for its low effect.

Inflammation is critical to determining MI size, and excessive inflammation leads to adverse left ventricular remodeling and heart failure [35]. While this is demonstrated well in the literature, our study did not find a difference in EF between overweight and non-overweight individuals.

Based on existing knowledge regarding the effect of obesity on the inflammatory response to insult, we assume this may partly explain the effect of BMI on mortality in STEMI patients suffering from AKI.

Other essential mechanisms linking BMI to ACS morbidity and mortality are the characteristics of obesity and independent patient factors. Denis et al. defined two clinical groups of patients with different susceptibility to cardiovascular disease, the normal weight with metabolic abnormalities which are prone to cardiac disease, and the obese metabolically healthy who do not suffer from excess morbidity and mortality; they were found to have a genetic difference which translated into a different inflammatory profile [36]. This phenomenon may be related to the type of obesity. Central obesity may be more hazardous in MI as it was associated with higher mortality in cardiovascular patients even with normal BMI [37]. Other studies found abdominal obesity represented by the waist and hip circumflex to be a stronger predictor of coronary disease than BMI [38,39].

There is conflicting evidence regarding the effect of BMI on infarct size and cardiac function. Analyzed data from six trials of patients undergoing PCI for STEMI assessed within one month with either cardiac magnetic resonance imaging (MRI) or ^99^mTc sestamibi single-photon emission computed tomography found no difference in infarct size, microvascular disease, or EF between higher and lower BMI [40]; on the other hand, several other studies found that infarct size was smaller [41,42] and EF was higher [42] in obese patients, yet not all patients in these cohorts suffered from STEMI, and only 45% received PCI [41]. In our cohort, overweight patients had a higher number of stenotic vessels. Yet, it did not translate into a worse outcome as their EF post PCI, and the incidence of hemodynamic instability was similar to non-obese. Further research is needed to conclude whether this may point to the possible protective effect on mortality.

It should not go unnoted that there is a rigorous debate in the literature on whether the obesity paradox is actual or merely a statistical phenomenon. Alternative explanations have been offered to describe why BMI modifies clinical risk. One argues that lower weight may reflect disease-related weight loss and poor nutritional status rather than health; both affect mortality risk. A major population-based cross-sectional study of almost 500,000 subjects found that being underweight was the most substantial independent risk factor for stroke and MI [43]. Our data did show that when BMI was divided into clinical classes, the largest proportion of deaths among patients was in BMI < 24.9. Our multivariant model showed that hemodynamic instability was a major risk contributor. A different view of our finding might be that mortality was due to direct cardiac injury, and reduced mortality may be explained by a better physiologic reserve in the elevated BMI group, representing a healthier group than the lower weight. Our research found a difference in mortality between BMI classes only in overall mortality and not in mortality at 30 days.

Another explanation may be that basic demographic variables also affect survival. A defined group of younger females was found in a study to be at a greater risk for in-hospital death after MI [44]; in our cohort, the AKI group had a higher percentage of females which had a higher prevalence of comorbidities, yet our survival analysis showed gender to be a major determinant of long-term but not short-term survival.

This study has several limitations. First, it is a retrospective analysis; our database was initially designed for a different primary goal, and therefore, was limited in its ability to adjust for all confounding variables and generalization of the results.

BMI in our cohort was calculated from self-reported weight and height. It is argued whether it is the most accurate index of obesity as body composition and waist circumference are argued to be more significant [22].

The univariant analysis included all subjects in the cohort, yet due to missing data, no recorded outcome, AKI or BMI, the multivariant analysis included fewer subjects which may have affected our results. Subgroup analysis is shown in Appendix A.

Our cohort included very few morbid obese and extreme underweight patients; therefore, these groups are under-represented compared to similar cohorts from literature.

Our follow-up did not include the change in BMI and its contribution to the primary outcome of overall mortality; therefore, it relates only to BMI at presentation and lacks the effect of change.

Our multivariant analysis found that hemodynamic instability was a significant risk factor for mortality and survival, with a difference in magnitude. There is a clinical association since decreased renal perfusion may cause AKI and affect survival. The presumed logic is that this short-lived event had a time-sensitive significance with an immediate affect and less on survival over time. No interaction or confounding was found between the two variables, yet the multivariant analysis may not fully control for it due to the limitation of our retrospective analysis.

## 5. Conclusions

We found that BMI has a protective effect on mortality among STEMI patients post PCI with AKI and that being overweight poses a survival benefit for these patients. This finding should not be read without considering that most of our cohort were overweight and class I obese.

This study did not attempt to investigate the mechanism behind this phenomenon, yet we proposed several hypotheses. Further research is needed to establish the pathophysiological effect of overweight on the natural course of STEMI and AKI.

## Figures and Tables

**Figure 1 jcm-11-06104-f001:**
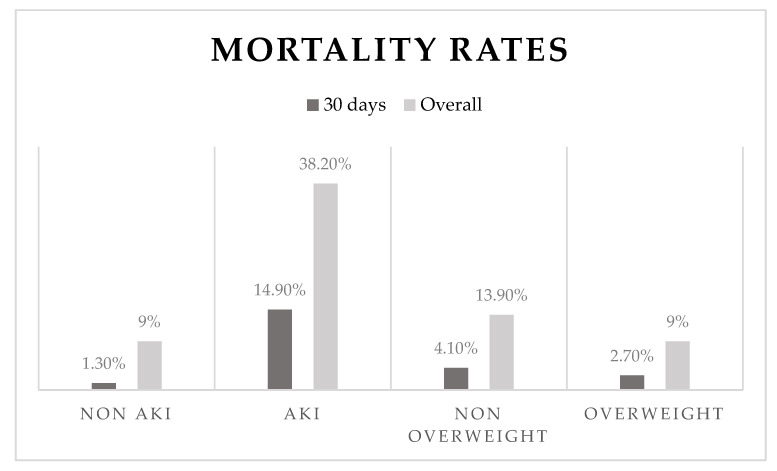
Mortality rates according to main effect.

**Table 1 jcm-11-06104-t001:** Baseline characteristics of patients who suffered and did not suffer from AKI.

	Patients without AKI	N	Patients with AKI	N	*p*-Value
N (1706)	1528 (89.6%)		178 (10.4%)		
Age, yrs	60.4 (±12.5)	1528	70.8 (±13.1)	178	<0.001
Gender (Female)	18.7%	1528	27%	178	0.009
BMI	26.80 (±4.0)	1093	27.1 (±4)	130	0.98
Smoker	52.1%	1528	33%	178	<0.001
DM	20.5%	1528	32%	178	<0.001
HTN	40.1%	1528	69%	178	<0.001
HPL	46.1%	1528	55%	178	0.20
Past MI	10.5%	1528	14.5%	178	0.09
Cr (mg/dL)	1.12 (±0.22)	1528	1.31 (±0.43)	178	<0.001
CRP (mg/L)	3.8 (1.3–10.1)	854	9.3 (2.3–20.4)	97	<0.001
CPK (U/L)	776 (313.5–1592.5)	1497	918 (331–2468)	172	0.04
Hb (g/dL)	14.4 (±1.5)	1526	13.60 (±1.8)	178	0.001
Time to reperfusion (min)	165 (105–390)	1527	232.5 (120–765)	178	0.001
Contrast volume (ml)	140 (115–175)	383	140 (97–155)	35	0.11
Hemodynamic instability *	2.20%	1527	16.90%	178	<0.001
CAD	0–0.2%, 1–45.1%, 2–29.9%, 3≤–24.8%	1524	0–0%, 1–36.5%, 2–27%, 3≤–36.5%	178	0.008
EF (%)	50 (40–55)	1505	45 (35–50)	176	<0.001
Follow-up time (wks)	45 (23–73)	1500	45 (23–71)	175	0.008

Values are mean ± SD, *n* (%), or median (interquartile range). * Hemodynamic instability was recorded as patients’ need for inotropes or intra-aortic balloon. CAD = number of vessels with significant coronary artery disease; CPK = creatine phosphokinase; Cr = creatinine on admission; CRP = C-reactive protein; DM = diabetes mellitus; EF = ejection fraction; Hb = hemoglobin; HPL = hyperlipidemia; HTN = hypertension; min = minutes; wks = weeks; yrs = years.

**Table 2 jcm-11-06104-t002:** Baseline characteristics of patients who are overweight (BMI > 25) and not overweight.

	Non-Overweight Patients	N	Overweight Patients	N	*p*-Value
	560 (35%)		1045 (65%)		
Age, yrs	63.7 (±13.6)	560	60.4 (±12.8)	1044	<0.001
Gender (Female)	41.90%	560	58.10%	1044	0.005
Smoker	53.90%	560	51%	1044	0.25
DM	17.10%	560	26.70%	1044	<0.001
HTN	37.90%	560	45%	1044	0.006
HPL	41.40%	560	49%	1044	0.004
Past MI	14.60%	560	14.60%	1044	0.9
Cr (mg/dL)	1.10 (±0.36)	559	1.10 (±0.33)	1042	0.74
CRP (mg/L)	3 (0.95–9.5)	419	5.4 (2.2–11.3)	783	<0.001
CPK (U/L)	885 (327–1707)	551	837 (341–1772)	1028	0.94
Hb (g/dL)	14.00 (±1.6)	559	14.40 (±1.5)	1043	<0.001
Time to reperfusion (min)	90 (110–390)	552	165 (105–420)	1024	0.36
Contrast volume (mL)	136.5 (110.5–168.5)	142	144.5 (117.7–176)	238	0.15
Hemodynamic instability *	4.50%	560	3.90%	1043	0.60
CAD	0–0.4%, 1–46.7%, 2–23.4%, 3–29.5%	555	0–0%, 1–40.2%, 2–32.2%, 3–27.6%	1035	0.001
EF (%)	45 (40–50)	556	45.90 (40–50)	1029	0.09
AKI	9.80%	429	11.10%	778	0.49
Follow-up time (wks)	46 (24–75)	545	45 (23–74)	1032	0.46

Values are mean ± SD, *n* (%), or median (interquartile range). * Hemodynamic instability was recorded as patients’ need for inotropes or intra-aortic balloon. AKI = acute kidney injury; CAD = number of vessels with significant coronary artery disease; CPK = phosphokinase; Cr = creatinine on admission; CRP = C-reactive protein; DM = diabetes mellitus; EF = ejection fraction; Hb = hemoglobin; HPL = hyperlipidemia; HTN = hypertension; min = minutes; wks = weeks; yrs = years.

**Table 3 jcm-11-06104-t003:** Mortality according to BMI class.

BMI Class	N	30-Day Mortality *p* = 0.17	N	Overall Mortality *p* = 0.02
<18.5	14	2 (14%)	14	4 (28.5%)
18.6–24.9	541	21 (4%)	546	74 (13.6%)
25–29.9	725	19 (2.5%)	730	68 (9.3%)
30–34.9	245	7 (3%)	246	21 (8.5%)
35–39.9	53	2 (3.5%)	56	4 (7%)
>40	12	0	12	1 (8.5%)

**Table 4 jcm-11-06104-t004:** Risk factors for mortality.

	30-Day Mortality N = 789	*p*-Value	Overall Mortality N = 797	*p*-Value
Age, yrs	1.08 (0.97–1.19)	0.13	1.07 (1.04–1.11)	<0.001
Gender (Female)	0.00 (0–0.005)	0.008	1.38 (0.64–2.95)	0.40
BMI	0.40 (0.21–0.75)	0.005	0.91 (0.84–0.99)	0.04
Smoker	1.51 (0.19–11.8)	0.69	2.65 (1.50–4.69)	0.001
DM	5.69 (0.52–61.36)	0.15	2.30 (0.85–6.19)	0.09
HTN	2.04 (0.34–12.30)	0.43	0.96 (0. 12–4.36)	0.96
HPL	19.46 (0.003–136,010.27)	0.51	0.74 (0.38–1.44)	0.39
Cr (mg/dL)	0.00 (0.00–0.00)	0.002	2.70 (0.002–4895.51)	0.79
CRP (mg/L)	1.06 (1.03–1.10)	<0.001	1.01 (1.00–1.02)	0.002
CPK (U/L)	1.0 (1.0–1.0)	0.10	1.00 (1.00–1.00)	0.93
HB (g/dL)	0.02 (0.002–0.22)	0.002	0.27 (0.07–1.03)	0.056
Time to reperfusion (min)	1.0 (0.99–1.002)	0.58	1.00 (0.99–1.00)	0.85
CAD	0.00 (0.00–0.07)	0.008	0.979 (0.59–1.62)	0.93
Hemodynamic instability *	247.02 (8.12–7512.77)	0.09	23.72 (5.19–108.26)	<0.001
EF (%)	0.82 (0.73–0.93)	0.002	0.61 (0.44–0.84)	0.003
AKI	41.51 (3.99–431.13)	0.002	2.91 (1.22–6.92)	0.01

Values are adjOR (CI95%). * Hemodynamic instability was recorded as patients’ need for inotropes or intra-aortic balloon. AKI = acute kidney injury; CAD = number of vessels with significant coronary artery disease; CPK = phosphokinase; Cr = creatinine on admission; CRP = C-reactive protein; DM = diabetes mellitus; EF = ejection fraction; Hb = hemoglobin; HPL = hyperlipidemia; HTN = hypertension; min = minutes; yrs = years.

**Table 5 jcm-11-06104-t005:** Risk factors for survival.

	30-Day Survival N = 756	*p*-Value	Overall Survival N = 781	*p*-Value
Age, yrs	1.23 (0.99–1.52)	0.05	1.06 (1.03–1.09)	<0.001
Gender (Female)	0.00 (0–0.001)	0.003	490.15 (20.03–11,994.34)	<0.001
BMI	0.77 (0.64–0.94)	0.01	0.93 (0.87–1.00)	0.054
Smoker	0.87 (0.19–3.84)	0.85	2.65 (1.50–4.69)	0.001
DM	7.17 (1.24–41.40)	0.02	28.67 (0.81–1009.09)	0.06
HTN	0.02 (0.00–4.08)	0.15	1.11 (0.62–1.96)	0.71
HPL	0.69 (0.19–2.49)	0.57	0.86 (0.48–1.51)	0.60
Cr (mg/dL)	0.00 (0.00–0.11)	0.03	1.54 (0.72–3.31)	0.26
CRP (mg/L)	1.01 (1.004–1.03)	0.009	1.01 (1.00–1.01)	<0.001
CPK (U/L)	1.0 (1.0–1.0)	0.28	1.00 (1.00–1.00)	0.90
HB (g/dL)	0.02 (0.002–0.18)	0.001	1.03 (0.85–1.24)	0.75
Time to reperfusion	1.0 (0.99–1.001)	0.98	1.00 (0.99–1.00)	0.16
CAD	0.76 (0.34–1.66)	0.49	1.06 (0.76–1.48)	0.71
Hemodynamic instability *	3.81 (0.78–18.54)	0.09	43.04 (0.14–12,402.22)	0.19
EF (%)	0.85 (0.79–0.92)	<0.001	1.01 (0.97–1.06)	0.43

Values are adjHR (CI95%). * Hemodynamic instability was recorded as patients’ need for inotropes or intra-aortic balloon. AKI = acute kidney injury; CAD = number of vessels with significant coronary artery disease; CPK = phosphokinase; Cr = creatinine on admission; CRP = C-reactive protein; DM = diabetes mellitus; EF = ejection fraction; Hb = hemoglobin; HPL = hyperlipidemia; HTN = hypertension; min = minutes; yrs = years.

## Data Availability

Not applicable.

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
