# Peer review of "BMI Modifies Increased Mortality Risk of Post-PCI STEMI Patients with AKI"

_jcm, 2022, doi:10.3390/jcm11206104_

Round 1
Reviewer 1 Report
First of all, I want to congratulate the authors to their laborious work. The manuscript deals with an interesting question, whether there is an interaction between acute kidney injury (AKI) and body-maas index (BMI) on survival after STEMI.
However several remarks have to be made:
2141 patients have been included in the study but only 1706 were analyzed for AKI and 1605 for BMI.
- Why were the other patients not analyzed?
- Is there a difference in baseline characteristics between patients analyzed or not analyzed?
- How many patients were analyzed only for AKI, how many only for BMI and how many patients were analyzed for both AKI and BMI?
- Would the results have been different, if only the patients, where both paramters (AKI and BMI) were available, were analyzed?
- Can the authors provide results for this group (both AKI and BMI are available)?
The number of patients analyzed for 30-day mortality is lower than the number of patients analyzed for 1-year mortality?? What is the reason?
The authors provide data on "mortality-risk" (Table 4) and "survival-risk" (Table 5).
- What is the difference between these two parameter apart from beeing reciprocal?
- Why are the significance levels for the various parameters different ?
Discussion section 2nd paragraph:
"in this study AKI incidence was not significantly higher in obese patients.". The difference in mortality between obese (11,1%) and non-obese patients (9,8%) was numerically 13%. Is the lack of significance of this difference due to the (rather) low number of patients compared with the literature?
Page 7, 7th paragraph:
"Yet, it did not translate into a worse outcome as their EF post PCI, and the incidence of hemodynamic instability was similar to non-obese, pointing to the possible protective effect from mortality." I think, this subgroup-analysis is probably premature.
Minor remarks:
- Introduction, 3rd paragraph: The first sentence is not gramatically correct and therefore not understandable
Table 3a: Giving the number of patients in the various groups (patients without AKI; patients with AKI; nn-Overweight patients; overweight patients) would make the table more readable
Author Response
1. Patients have been included in the study-
Thank you for this comment- here are the explanations and also revisions performed:
- Why were the other patients not analyzed? This retrospective analysis was performed on electronic medical records and many data were missing including outcome and the core variants AKI diagnosis and BMI . Patients without these core variant were not included in the final analysis leading to a smaller number.
- Is there a difference in baseline characteristics between patients analyzed or not analyzed? We have performed subgroup analysis for these patients and added as appendix A. Further discussion was also added to the limitation paragraph.
- How many patients were analyzed only for AKI, how many only for BMI and how many patients were analyzed for both AKI and BMI?
1706 had a record for AKI diagnosis yes/no
1605 had a recorded BMI
of these not all had other baseline variants yet a baseline analysis was performed for all.
final multivariant analysis was performed on
N=789 for 30 days mortality
N=797 for overall mortality
N=756 for survival at 30 days
N= 781 for overall survival
The number of patient for each univariant analysis was added to the tables.
- Would the results have been different, if only the patients, where both paramters (AKI and BMI) were available, were analyzed?
Our results describe only patients with both variants.
We assume the missing patients may have effected our results, we will further discuss this in the limitation paragraph.
- Can the authors provide results for this group (both AKI and BMI are available)? We presented results only for patients with both recorded AKI and BMI. As these variants are at core of our research we did not look at outcome in patients without these recorded variants.
2. The number of patients analyzed for 30-day mortality is lower than the number of patients analyzed for 1-year mortality?? What is the reason?
We had missing data at 30 days as not all patients had a recoded visit yet more more recoded for the rest of the follow up leading to the difference in numbers.
3. The authors provide data on "mortality-risk" (Table 4) and "survival-risk" (Table 5).
- What is the difference between these two parameter apart from being reciprocal? We performed a cox analysis in addition to a logistic regression in-order to control and include the impact of the stringent follow up time performed .
- Why are the significance levels for the various parameters different ? There is a difference between the effect of variants on mortality and survival, mostly speaking, regarding hemodynamic instability and gender. We believe instability to be a short-lived event with a time sensitive effect which translate into a different significance. This was further explained in the discussion section. Gender had a different effect in concordance to known literature.
4. Discussion section 2nd paragraph: Our cohort included mostly overweight and class I obese, yet most cases with AKI were in the normal weight and overweight class, we believe this has influenced our results. The frequencies of BMI classes is added to appendix A.
5. Page 7, 7th paragraph- thank you for this comment, a more cautious wording has been used.
6. Introduction, 3rd paragraph: thank you for this comment, the wording has been clarified.
7. Table 3a: The were different numbers analyzed for each variant due to missing data, the table has been converted to a figure due to the academic advisors comments in-order to make it more intuitive.
Reviewer 2 Report
This is a clinically relevant analysis describing the impact of BMI on outcome of patients with STEMI and AKI. The study is generally well written and the methodology of the study is appropriate.
Major comments:
- The study conclusions are inappropriate. Should be rephrased to include the findings of the study in relation to the aim of the study. This study did not assess the effect of being overweight on both short- and long-term inflammatory response to cardiac injury
- The discussion section does not discuss appropriately the interaction between BMI and AKI and the potential mechanism which led to the protective effect of BMI on mortality in AKI pts. Inflammation is a hallmark of STEMI, AKI and BMI and it is not clear what the hypothesis is. Moreover, this study only assessed CRP levels and no other more specific inflammatory test.
Minor comments:
- Spelling, grammar and English writing should be double checked – Some examples – Line 36 – PC, Line 40 BMI is a strongly predicts, Lines 98-100 are left over from the article template, line 113
- Lines 66 – 68 are better suited for the statistical analysis chapter
- Line 191 – should be rephrased – “obesity… causes vasoconstriction leading to glomerulopathy” is too general a statement.
- Tables 4 and 5 headings are misleading. It is not clear what they display. The number of patients is different in the analysis. The table should be self-explanatory without the above text
- A drawback could be the assessment of only self-reported BMI and not actual measurements of Height, weight, waist circumference or othe measurements of body fat.
- Maybe the interaction of renal function (not only AKI) with BMI in STEMI patients can be discussed – example this paper 10.3389/fcvm.2021.765153
Author Response
Thank you for your comments, we have answered and revised according to your suggestions, the following sections
- The study conclusions- the section has been revised.
- The discussion section does not discuss appropriately the interaction between BMI and AKI- the section has been revised
- Spelling- we have rechecked for spelling and phrasing.
- Lines 66 – 68- Have been moved as recommended.
- Line 191- has been rephrased as recommended.
- Tables 4 and 5 headings are misleading. It is not clear what they display. The number of patients is different in the analysis.The table should be self-explanatory without the above text.
Patient numbers differ as some data of mortality or BMI is missing. We have tried to emphasize the denominator where critical and leave out where it is not to prevent overloading the tables. - A drawback could be the assessment of only self-reported BMI and not actual measurements of Height, weight, waist circumference or othe measurements of body fat-
We have extended the paragraph discussing this issue. - Maybe the interaction of renal function (not only AKI) with BMI in STEMI patients can be discussed – example this paper 10.3389/fcvm.2021.765153
Thank you, the study results and methodology have been added.